# Factors Controlling Dead Wood Decomposition in an Old-Growth Temperate Forest in Central Europe

**DOI:** 10.3390/jof8070673

**Published:** 2022-06-27

**Authors:** Mayuko Jomura, Riki Yoshida, Lenka Michalčíková, Vojtěch Tláskal, Petr Baldrian

**Affiliations:** 1Department of Forest Science and Resources, College of Bioresource Sciences, Nihon University, Fujisawa 252-0880, Kanagawa, Japan; majomura@gmail.com; 2Laboratory of Environmental Microbiology, Institute of Microbiology of the Czech Academy of Sciences, 142-20 Prague, Czech Republic; lenka.michalcikova@biomed.cas.cz (L.M.); tlaskal@biomed.cas.cz (V.T.); baldrian@biomed.cas.cz (P.B.); 3Department of Biology, Faculty of Science, University of Hradec Kralove, Hradecká 1285, 500-03 Hradec Králové, Czech Republic

**Keywords:** respiration, fungal community, fungal biomass, extracellular enzymes, chemical properties, structural equation modeling

## Abstract

Dead wood represents an important pool of carbon and nitrogen in forest ecosystems. This source of soil organic matter has diverse ecosystem functions that include, among others, carbon and nitrogen cycling. However, information is limited on how deadwood properties such as chemical composition, decomposer abundance, community composition, and age correlate and affect decomposition rate. Here, we targeted coarse dead wood of beech, spruce, and fir, namely snags and tree trunks (logs) in an old-growth temperate forest in central Europe; measured their decomposition rate as CO_2_ production in situ; and analyzed their relationships with other measured variables. Respiration rate of dead wood showed strong positive correlation with acid phosphatase activity and negative correlation with lignin content. Fungal biomass (ergosterol content) and moisture content were additional predictors. Our results indicate that dead wood traits, including tree species, age, and position (downed/standing), affected dead wood chemical properties, microbial biomass, moisture condition, and enzyme activity through changes in fungal communities and ultimately influenced the decomposition rate of dead wood.

## 1. Introduction

Dead wood is one of the main stocks of carbon and nitrogen in forest ecosystems and represents a significant source of forest carbon and nitrogen cycles through decomposition processes [1]. Dead wood supports many decomposer organisms by supplying nutrients and habitat. It hosts diverse fungi, bacteria, and invertebrates, which are the bottom of the detritus food chain in forests [2]. As decomposition proceeds, carbon in dead wood becomes sequestered in soil for thousands of years [3]. Atmospheric nitrogen fixed by bacteria and soil organic nitrogen mineralized by fungi and immobilized, as fungal biomass in dead wood increase the availability of nitrogen and carbon in forest ecosystems [4,5,6]. Therefore, a quantitative understanding of carbon and nitrogen cycling through dead wood decomposition is indispensable to estimate the carbon sequestration ability of forests and the maintenance of biodiversity in forest ecosystems.

The dead wood decomposition rate has been widely analyzed and reported on for different forest types, and the influences of tree species [7,8], dead wood position [9,10], decay age [11,12], climatic factors [13], chemical properties such as nitrogen and lignin content [14], enzyme activities, decomposer biomass [15], and fungal community structure [16] on dead wood decomposition rate have been determined in the field or at the experimental level. However, it is widely acknowledged that these factors have correlations and cause–effect relationships with each other under different time and scale conditions, and the complexity of these interactions has prevented us from completely understanding the process of dead wood decomposition. 

The wood from different tree species has various chemical and physical properties [17]. Conifers are known to have higher lignin content than hardwood species, and heartwood is known to contain higher amounts of secondary metabolites (phenolics and tannins) than sapwood, which affects microbial biomass, colonization, and decomposition rates over the long term [14,18,19]. Tree species also have an impact on the assembly history through differences in the endophyte community, which represents the earliest decomposers of dead wood [20,21]. Tree position at senescence (standing snags and downed logs) influences environmental conditions such as temperature and moisture, which are the main controllers of microbial biomass, colonization, and decomposition rate [22,23]. The position of dead wood also controls access to the ground for nutrient uptake and as a source of microorganisms. Decay length affects succession stages of decomposers, and long-term chemical degradation tends to increase relative contents of recalcitrant compounds such as lignin. Microbial community structure at certain time points reflects the assembly history and microbial succession in accordance with tree species, decay ages, and microbial interactions [16,24,25,26,27,28], but the rate of the succession is controlled not only by decay length but also by fungal colonization and decomposition rate. 

Enzymes are direct mediators of wood decomposition under the strong control of environmental factors, microbial community, and dead wood chemical properties. High activity of hydrolytic and oxidative enzymes, which is related to holocellulose and lignin degradation, leads to higher decomposition rates [29,30,31]. However, the effect of enzyme activities on decomposition rate are not simple [7,32], presumably because of the differences in contribution of individual enzymes, microbial species [33,34], temperatures, water conditions and pH [33,35,36,37,38], microbial interactions [39,40,41], and trade-off effects between enzyme production and decomposition [42]. Additionally, enzyme activity and decomposition rate observed weight and density loss are less compatible because of differences in time scale.

To elucidate the complicated cause-effect structure, including directional influences, between possible controlling factors of the decomposition rate of dead wood, we examined dead wood for which the tree species, positions, and decay length were known in a temperate old-growth forest in central Europe. The aims of this study were to determine (1) how physico-chemical and environmental factors, fungal community composition, and enzyme activities are associated with tree species, positions, and decay length of dead wood and (2) how dead wood respiration rate is explained by the cause–effect relationships of possible factors with direct and indirect effects on dead wood decomposition rate.

## 2. Methods

### 2.1. Study Site

The study site was located in the Žofínský Prales National Nature Reserve (Zofin Forest), an unmanaged old-growth forest in the south of the Czech Republic. Parts of the forest have not been managed since 1838, when it was declared a reserve. The study area is situated at 730 to 830 m above sea level, and the subsoil is almost homogeneous and consists of fine- to medium-grain porphyritic and biotite granite. Annual mean precipitation is 866 mm, and annual mean air temperature is 6.2 °C [43]. This forest is covered mainly by *Fagus sylvatica* (51.5% of total living biomass), *Picea abies* (42.8%), and *Abies alba* (4.8%) trees. All the living and dead trees have been monitored since 1975 in a 25 ha core zone of the reserve, which is currently included in the global network, ForestGEO [43,44]. A circular plot with a 30 m radius was established in the center of the 25 ha plot (48.666 N, 14.707 E). There were 51 dead wood objects (coarse woody debris and tree trunks with diameter >10 cm) in the circular plot, including three tree species (beech, spruce, and fir), two positions (standing and downed dead wood), and four decay lengths. 

### 2.2. Respiration Measurement, Sampling of Dead Wood, and Sample Processing

Respiration measurement was conducted in situ on 10 and 11 October 2019, with an infrared gas analyzer (IRGA, GMP343, Vaisala Inc., Vantaa, Finland) and an acrylic chamber (diameter: 9 cm, height: 8 cm). Two vinyl chloride collars were fixed on the dead wood with putty to seal the gap between the collar and the wood (Figure 1). An acrylic chamber equipped with the IRGA was placed on the collar for 5 min, and CO_2_ concentration was recorded every second by a data logger (GL200A, Graphtech Inc., Yokohama, Japan). Temperature and humidity in the chambers were also recorded. After respiration measurement, the temperature of the wood at a depth of 3 cm was measured with a thermometer. Two wood blocks (ca. width: 3 cm, depth: 3 cm, length: 10 cm) were taken from under the respiration measurement point using a hammer and chisel, and half of the blocks were kept in air temperature, and the other half of the samples were stored frozen until the next stage of sample processing.

The respiration rate of the dead wood blocks kept in air temperature was measured in a laboratory using an acrylic chamber (12 × 12 × 20 cm) one day after sampling to assess any differences in respiration rate between measured conditions in situ and in the laboratory. The procedures used to measure the blocks were the same as those used in the field. The wet weight of the wood blocks was measured; the blocks were then freeze-dried and weighed again to calculate their water content. Their size was also recorded to calculate the wood density of the freeze-dried block. The blocks were then ground to a fine powder with a mill (Ultra Centrifugal Mill ZM 200, Retsch, Haan, Germany) for the subsequent analyses. The other half of the blocks were also freeze-dried and ground to a fine powder with the same mill for the following DNA extraction.

### 2.3. Enzyme and Chemical Analyses

Enzyme activities of endocellulase, cellobiohydrolase (exocellulase), β-glucosidase, endoxylanase, β-xylosidase, β-galactosidase, α-glucosidase, *N*-acetylglucosaminidase, phosphomonoesterase (phosphatase), esterase (lipase), laccase, and Mn-peroxidase were measured as described in a previous study [45].

The amount of Klason lignin in the sample was estimated gravimetrically by using hot sulfuric acid digestion with a sample degreased by organic solvent (ethanol-toluene, 1:2) [46]. Sugar content was estimated by the phenol-sulfuric acid method [47]. Hollocellulose content was calculated by subtracting the lignin and holocellulose content from the dry mass of the degreased sample.

### 2.4. DNA Extraction and Amplification

Total genomic DNA was extracted from 200 mg of wood powder using the NucleoSpin Soil Kit (Macherey-Nagel, Duren, Germany) according to the procedure described in Baldrian et al. [45]. Extracted DNA was used as a template for the amplification of the fungal ITS2 region using barcoded primers gITS7 and ITS4 [48] and the bacterial V4 region using barcoded primers 515F and 806R [49] in three PCR reactions per sample as described previously [50]. PCR reactions contained 2.5 μL of 10× buffer for DyNAzyme DNA Polymerase, 0.75 μL DyNAzyme II DNA polymerase (2 u μL^−1^), 1.5 μL of BSA (10 mg mL^−1^), 0.5 μL of PCR Nucleotide Mix (10 mM), 1 μL of each primer (10 μM), 1.0 μL of template DNA (concentration approximately 5–50 ng μL^−1^), and sterile ddH2O up to 25 μL. Conditions for amplification started at 94 °C for 4 min followed by 35 cycles of 94 °C for 45 s, 50 °C for 60 s, and 72 °C for 75 s and finished with a final setting of 72 °C for 10 min. Amplicons were purified, pooled, and sequenced on the Illumina MiSeq to obtain pair-end sequences of 2 × 250 bp.

Fungal and bacterial rRNA gene copies were quantified by qPCR using FR1 and FF390 primers for fungi [51] and 1108f and 1132r primers for bacteria [52,53] as described previously [50]. Conditions for amplification started at 56 °C for 2 min and 95 °C for 10 min followed by 40 cycles of 95 °C for 15 s, 50 °C for 30 s, and 70 °C for 1 min and finished with a final setting of 95 °C for 10 min.

### 2.5. Bioinformatic Workflow

Sequence data processing was performed using pipeline SEED 2.0 [54] as described in Baldrian et al. [45]. Briefly, paired-end reads were merged using fastq-join [55]. The ITS2 region was extracted with ITS Extractor 1.0.8 [56] before processing. Chimeric sequences were detected and deleted using Usearch 7.0.1090 [57], and sequences were clustered at a similarity level of 97% using UPARSE implemented in USEARCH [58]. Consensus sequences were generated for each cluster, and the closest hits at the species level were identified using BLASTn against UNITE [59] and GenBank. In cases where the best hit showed a similarity of less than 97% with 95% coverage, the best genus-level hit was identified. The species-level analyses were performed on a dataset where operational taxonomic units (OTUs) belonging to the same species were combined, and all other OTUs were combined into the genus of the best hit and designated “sp.” Sequences identified as nonfungal were discarded. Sequence data were deposited in NCBI database under the accession number PRJNA827576. To assign putative ecological functions to the fungal OTUs, the fungal genera of the best hit were classified into ecophysiological categories (e.g., white-rot, brown-rot, saprotroph, and ectomycorrhiza) based on the published literature [60]. The definition of categories was the same as in [60]. Fungal OTUs not assigned to a genus with known ecophysiology and those assigned to genera with unclear ecophysiology remained unassigned.

### 2.6. Statistical Analysis

Statistical analysis was performed with R software [61]. One-way ANOVA was conducted to assess the effect of tree species, dead wood position, and decay length on observed variables. The Student’s *t*-test was used for correlation analysis among the observed variables. Two-dimensional non-metric multidimensional scaling (NMDS) ordination analysis on Bray–Curtis distances of fungal and bacterial community was performed in R using the *vegan* package [62].

The influence of all physico-chemical and biological factors on respiration rate was analyzed by using a generalized linear model (GLM). To eliminate the correlation effect on GLM analysis, we selected moisture content, lignin content, bacterial copy number, ergosterol content, OTU richness from physico-chemical and biological factors, brown rot, white rot, other saprotrophs, and ectomycorrhizal from fungal ecological groups and acid phosphatase, chitinase, β-galactosidase, α-glucosidase, and endo xylanase in enzyme activity as controlling factors of respiration rate of dead wood in situ. All the factors were log-transformed and z-scored.

To assess cause–effect relationships surrounding respiration rate, structural equation modeling (SEM) was conducted using the *lavaan* package [63]. Factors selected in the GLM analysis were used; fungal NMDS axes 1 and 2 and brown rot abundance were also accepted in the model.

## 3. Results

### 3.1. Dead Wood Physico-Chemical and Biological Properties and Respiration Rate

Dead wood density decreased with decay length, and snag wood density was higher than log density for all species (Figure 2). Lignin content increased with decay length (*p* < 0.05), reaching more than 60% of relative content in coniferous snags and logs that had been decaying for 22–43 years. Moisture content of dead wood was significantly lower in snags for all species (*p* < 0.01). Fir and spruce showed significantly lower pH than beech (*p* < 0.01). Ergosterol content decreased as decay length increased (species, *p* < 0.01; decay length, *p* < 0.01), and beech had the highest content of the three tree species. Copy numbers of bacteria were significantly different among species (*p* < 0.05) and positions (*p* < 0.01), whereas fungal copy numbers showed no significant differences in the dead wood categories. Acid phosphatase activity differed among species (*p* < 0.05), and snags had lower activities than logs (*p* < 0.01). The dead wood categories were not significantly different in terms of fungal OTU richness or Shannon–Wiener diversity index. The respiration rate of dead wood significantly decreased as decay length increased (*p* < 0.05). 

### 3.2. Microbial Community Composition and Ecological Traits

In total, 532,265 fungal sequences were obtained and clustered into 9853 OTUs. After removal of non-fungal sequences and singletons, there were 3745 OTUs. Taxonomic evaluation revealed that most of the OTUs belong to the phylum Ascomycota and Basidiomycota (Figure 3). There were different ratios in the dead wood categories. Fir snags and logs and young spruce logs had more Basidiomycota, whereas spruce snags and old logs and beech snags and logs had much more Ascomycota than Basidiomycota. The most abundant genera were *Protounguicularia*, *Spadicoides*, *Cystostereum*, *Fomitopsis*, *Hyphodonitia*, *Ischnoderma*, and *Xylodon*, with different ratios among dead wood categories. Brown rot emerged in snags of all species and decreased with decay years in spruce logs. In total, 569,976 bacterial sequences were clustered into 22,363 OTUs, and 7160 OTUs were finally obtained after removal of singletons. The most abundant taxa belonged to the *Acidobacteria*, *Actinobacteria*, and *Alphaproteobacteria*; these taxa occupied more than half of the total abundance of each of the dead wood categories (Figure 3).

### 3.3. Correlations among Dead Wood Respiration Rate and Its Potential Controlling Factors and GLM Results

There was no correlation between the respiration rate of dead wood measured in situ and in the laboratory (Figure 4). Most enzyme activities were positively correlated to each other and laccase, and four hydrolytic enzymes had significant positive correlations with dead wood respiration rate in situ. Relative abundance of fungal ecological groups did not show any significant correlations with dead wood respiration rate in situ except for fungal NMDS axis 2. Fungal biomass (both ergosterol content and rDNA copy numbers) showed strong correlations with dead wood respiration rate in situ, and ergosterol content was strongly correlated to hydrolytic enzyme activities. Cellulose and lignin content had a strong negative correlation, and they showed many positive and negative correlations with the other measured variables.

The factors selected for the final GLM were lignin content (*p* < 0.01) and acid phosphatase activity (*p* < 0.01), and the model explained 31% of the variation in respiration rate in situ. Acid phosphatase and lignin content were significantly positively and negatively correlated, respectively, to the respiration rate (Figure 4).

### 3.4. Structural Relationships between Dead Wood Properties, Fungal Community Structure, Enzyme Activities, and Dead Wood Respiration Rate

SEM analysis showed direct relationships between dead wood traits, physico-chemical and microbial properties, enzyme activities, and dead wood respiration rate in situ (Figure 5). Respiration rate was directly controlled by acid phosphatase and lignin content. Acid phosphatase was affected by ergosterol content, moisture content, and fungal NMDS axis 1. Ergosterol content was negatively controlled by decay length and lignin content. Lignin content was controlled by fungal community composition (NMDS axis 2), which was controlled by decay length. Dead wood species affected fungal community composition (NMDS axis 1) and ergosterol content. Dead wood position affected moisture content and fungal community composition (NMDS axis 1). The other enzyme activities, fungal OTU richness, bacterial abundance, and bacterial communities were not adopted in the model. The comparative fit index (CFI), Tucker–Lewis index (TLI), relative root mean-square error of approximation (RMSEA), and *p*-value of the model were 0.988, 0.983, 0.035, and 0.566, respectively, and the model explained 30% of the variation in respiration rate. 

## 4. Discussion

### 4.1. Respiration Measured In Situ and in the Laboratory

We conducted two respiration measurements under in situ and laboratory conditions. Even though laboratory measurements were conducted only one day after the in situ respiration measurements, no significant correlation was observed between them. This means that respiration measurements of dead wood cut samples does not reflect the decomposition rate in the field. Therefore, all of the analyses reported in this study used in situ respiration data.

### 4.2. Enzyme Activities and Respiration Rate

Although there is a limited body of literature, clear positive correlations between oxidative and hydrolytic enzyme activities and mass loss based decomposition rate of dead wood have been observed in several studies [7,30]. These data, however, only partly address the dependence of decomposition rates on enzyme activity because decomposition proceeds over long time periods, whereas enzyme activities are only measured at certain points in time. To directly link decomposition rates with their predictors, including enzyme activity, it is necessary to estimate them simultaneously. The present study was the first attempt to explore these relationships. 

We measured two oxidative and ten hydrolytic enzyme activities, and significant positive correlations with respiration rate were observed for five hydrolytic enzymes: α-glucosidase, β-galactosidase, chitinase, endo xylanase, and acid phosphatase. 

GLM analysis selected acid phosphatase from these five enzymes as a controlling variable of respiration rate, and SEM analysis also revealed acid phosphatase was a direct controlling factor of respiration rate. This suggests that acid phosphatase activity is strongly linked to the respiration rate of dead wood, probably because acid phosphatase is the main enzyme in phosphorus acquisition. The result is consistent with those of previous reports, which showed that acid phosphatase positively correlates with the decomposition rate of woody material and soil organic matter [30,64]. 

Acid phosphatase was controlled by ergosterol, which had the largest standardized coefficient, suggesting that fungal biomass is the strongest controlling factor of acid phosphatase activity. This result is consistent with those of previous reports showing a positive correlation between enzyme activity and fungal biomass [32,45]. 

The high acid phosphatase activity caused by high dead wood moisture content was controlled by the dead wood position in this study. Acid phosphatase has been reported to increase with increasing water content [36,37,64,65,66]. These results suggest that the dead wood log position increased moisture content, leading to higher acid phosphatase activity, and resulting in an increased respiration rate. It is generally acknowledged that downed logs decompose faster than snags [9,10]. Jomura et al. [67] also found that the low respiration rate of snags was caused by a relatively low moisture content (20% lower than logs). Therefore, the slow decomposition rate of snags can be explained by low enzyme activities resulting from relatively lower moisture conditions.

According to the SEM model, acid phosphatase was also controlled by dead wood position through fungal communities. Because fungal NMDS axis 1 was associated with snag position and brown rot dominance, fungal communities specified by dead wood position had a negative effect on acid phosphatase activity. This result suggests that fungal communities specific to snags affect acid phosphatase activity separately from moisture content. Snags have less contact with soil, which is the main phosphorus supplier, so the phosphorus content in snags is expected to be lower than that of logs. The relative lack of phosphorus in snags will result in these specific fungal communities and ultimately cause lower levels of acid phosphatase activity. 

### 4.3. Other Factors Potentially Affecting Deadwood Respiration Rate

Lignin content was another factor showing a direct negative effect on respiration rate. The SEM model using chitinase activity, which was correlated to respiration rate and had a negative correlation to lignin content as a mediating factor of the path from lignin to respiration rate, was not adopted. The other SEM model which latent variable was added in the path between lignin content and respiration rate with the expectation that there were unmeasured variables associated with the lignin effect on respiration was not also adopted. These results suggested that the negative effect of lignin content on respiration rate could be explained by the lignin content itself. In industrial biofuel production, there is considerable evidence that lignin reduces hydrolytic enzyme efficiency in pretreated woody material (see [68]). Moreover, lignin preferentially binds at the same sites preferred by cellulase, inhibiting cellulose degradation [68]. Under natural conditions, an increased lignin/N ratio decreases the decomposition rate of leaf litter [14]. Therefore, high lignin content should decrease the decomposition rate of dead wood because it inhibits the accessibility and the binding efficiency of hydrolytic enzymes to the target substrate.

Lignin content was not controlled by decay length, but it was affected by fungal communities associated with fungal NMDS axis 2 in the SEM model. Previously, an increase in lignin content with increasing decay length or decreasing wood density has been reported [69,70], and white rot fungi are also known to decrease lignin content [71] as indicated by the significant negative correlation in our study. The results of this study may indicate fungal community composition may be more important than decay length as a controlling factor of lignin content.

Ergosterol content was controlled by dead wood species, decay length, and lignin content. We observed higher content of fungal biomass in beech than in spruce and fir, which is consistent with previous results showing that decaying angiosperms have higher fungal biomass than gymnosperms [18,19,72]. This also explains the faster decomposition rate of angiosperm dead wood than gymnosperm dead wood, which has been widely acknowledged [8].

Fungal communities were controlled by all dead wood traits, and there were two main pathways to respiration rate in the SEM model. Nevertheless, fungal communities showed no direct effect on respiration rate in terms of fungal OTU richness or the diversity index. This result is probably due to the fact that the DNA-based description of fungal community composition also included non-active groups. Fungal community indirectly controlled the respiration rate through the changing chemical and biological conditions of dead wood.

## 5. Conclusions

Dead wood decomposition in our study area proceeded from 7 to more than 43 years along with the increase of lignin content through the activities of dead wood-associated microorganisms. The microbial communities and biomass were controlled by dead wood species and position. This study elucidated the structure of the interactions among these factors and the direct controlling factor of the dead wood respiration rate. We were able to determine the complex interactions because the history of the dead wood had been monitored over the long term, and we studied dead wood with known duration of decay and tree species. Dead wood is an important component in the forest carbon budget and plays a role in nitrogen cycling via nitrogen fixation and immobilization. Long-term monitoring of dead wood dynamics is essential to determine material cycling in forest ecosystems.

## Figures and Tables

**Figure 1 jof-08-00673-f001:**
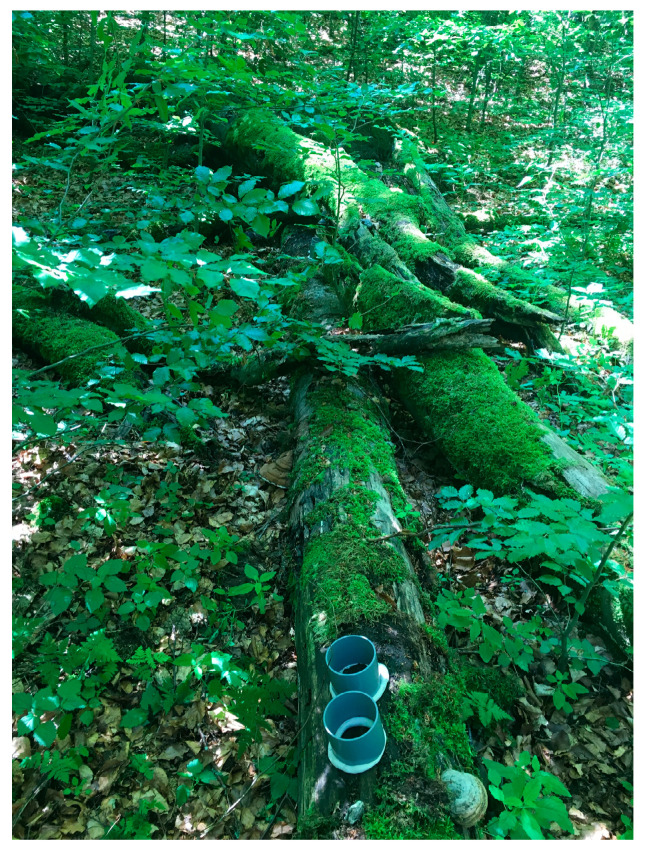
Beech downed dead wood with measurement collars in Zofin Forest in central Europe.

**Figure 2 jof-08-00673-f002:**
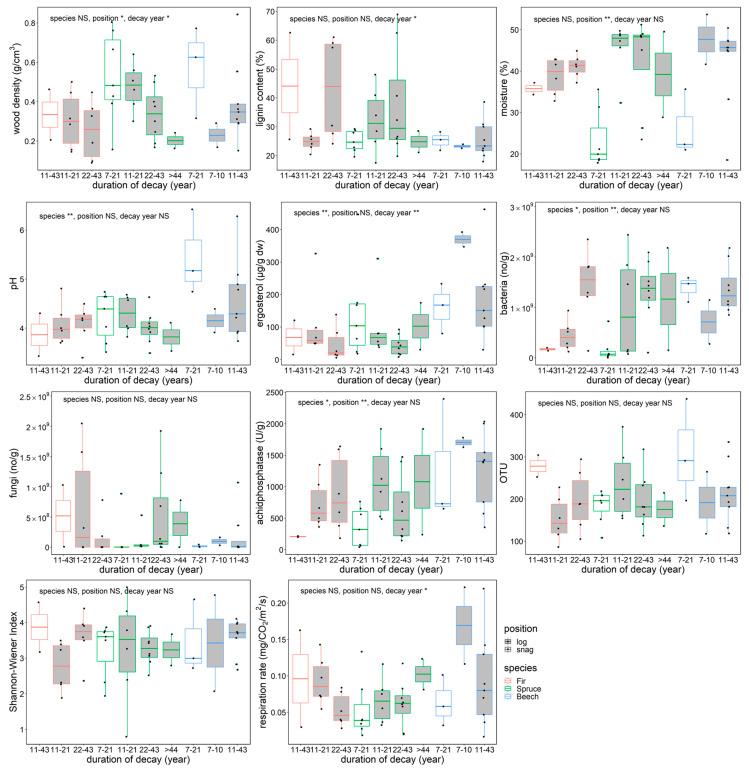
Physico–chemical and biological features and in situ respiration rate of beech, spruce, and fir dead wood snags and logs in an old-growth forest in the Zofin Forest by species and duration of decay. The significance of the effect of tree species, dead wood position, and decay length are indicated with asterisks (ANOVA, * *p* < 0.05, ** *p* < 0.01).

**Figure 3 jof-08-00673-f003:**
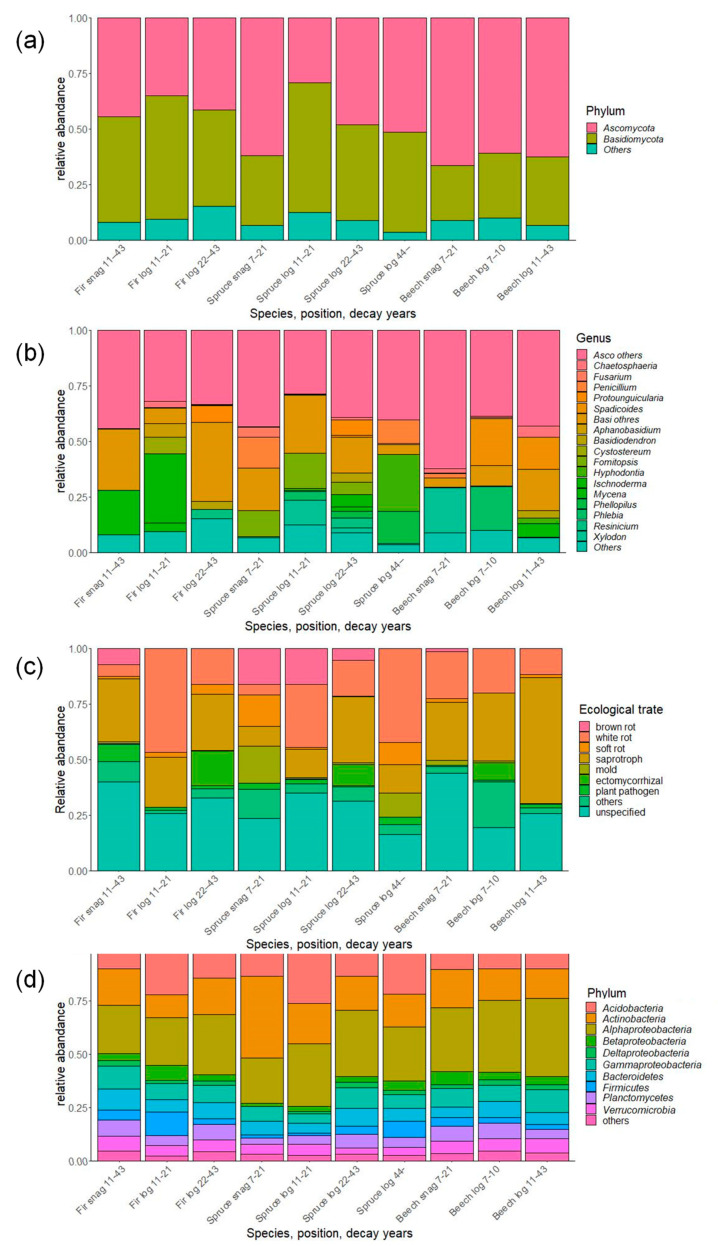
Composition of fungal and bacterial communities associated with dead wood in the Zofin Forest on the level of (**a**) fungal phylum, (**b**) genus, (**c**) ecological trait, and (**d**) bacterial phylum/class.

**Figure 4 jof-08-00673-f004:**
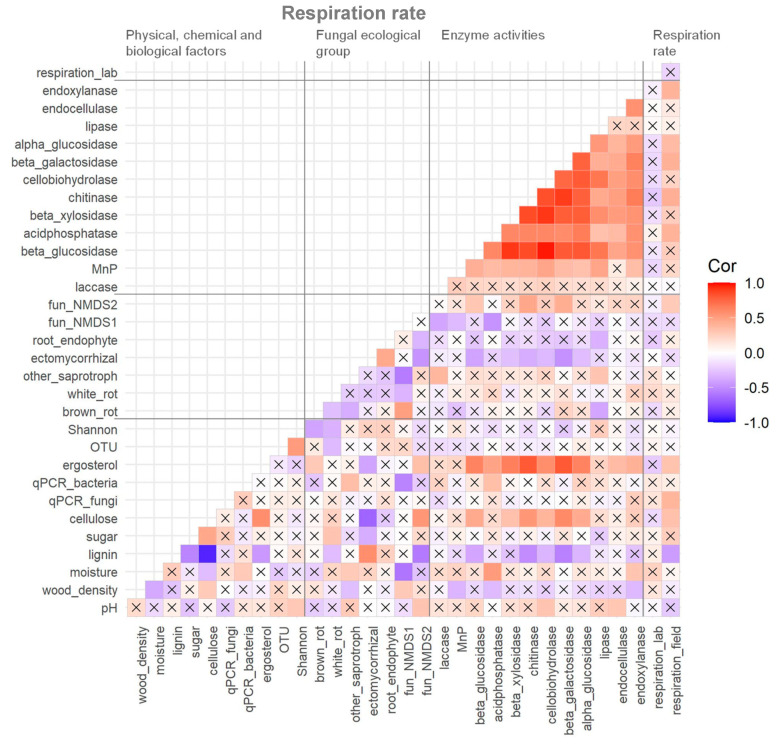
Correlation matrix of measured variables of physico–chemical and biological factors, relative abundance of fungal ecological groups, enzyme activities, and respiration rate of dead wood in the Zofin Forest. The cross marks indicate the absence of significant correlation (*p* < 0.05).

**Figure 5 jof-08-00673-f005:**
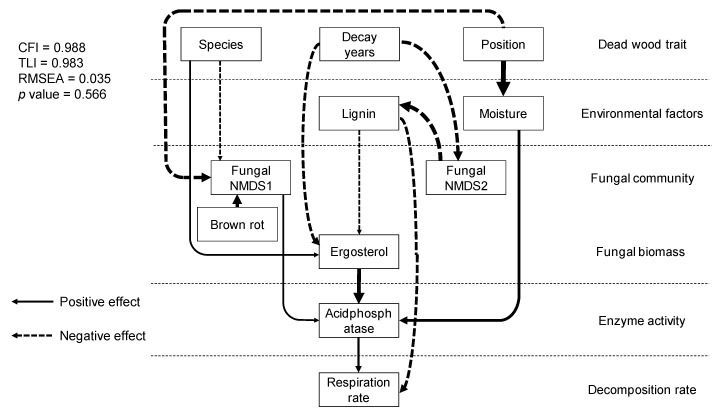
Structural relationships between dead wood properties, physico-chemical and biological factors, fungal community structure, enzyme activity, and in situ dead wood respiration rate. The lines and dotted arrows indicate positive and negative relationships between factors, respectively. The arrow width indicates standardized path coefficients. All of the paths shown are significant (*p* < 0.05).

## Data Availability

The data presented in this study are available within the article and the Appendix A.

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
