# Peer review of "Factors Controlling Dead Wood Decomposition in an Old-Growth Temperate Forest in Central Europe"

_jof, 2022, doi:10.3390/jof8070673_

Round 1

Reviewer 1 Report

This article deals with the factors related to the decomposition of dead wood and the correlation between the factors. Although many studies have been conducted to find the factors that cause the decomposition of dead wood, I think this is a very meaningful study because studies that explain the relationship between causes are rare. However, I suggest some minor changes for publishing this article.

1) Indicate the units in the Fig 1 accurately.

2) Correct the subheading number of Materials and Methods (3.2 duplicate use).

Author Response

Comments to reviewer #1

1) Indicate the units in the Fig 1 accurately.

 We changed the units of wood density and respiration rate in an appropriate style.

2) Correct the subheading number of Materials and Methods (3.2 duplicate use).

 We corrected the subheading number.

We have revised our manuscript to address the other reviewers’ comments. We would like to add raw data for this study as a supplementary table for secondary use of the data. We made minor corrections as it follows.

L42      “N” was modified to “nitrogen”.

L91      “1977” was modified to “1975”.

L98      Date that measurement was conducted was added.

L100    Size (diameter and height) of the acrylic chamber was added.

L101    “Figure 1” was inserted.

L129    “degreased sample” was deleted.

L131    “degreased” was added to specify the sample condition.

L192    Figure number was modified.

L196    P value for species was modified.

L206    Figure number was modified.

L213    Figure number was modified.

L220    Name of phylum of bacteria was italicized.

L222    Figure number was modified.

L223    Figures were made bigger and rearranged. Name of phylum of bacteria in (d) was italicized.

L225    Figure number was modified.

L230    Figure number was modified.

L231    “but” was modified to “and”.

L241    Figure number was modified.

L247    Figure number was modified.

L252    Figure number was modified.

L266    Figure number was modified.

L269    “E=error variance” was deleted.

L380    Number of authors was shortened to ten.

L393    Page number was corrected.

L396    Number of authors was shortened to ten.

L398    Number of authors was shortened to ten.

L425    Doi was corrected.

L432    Page number was corrected.

L436    Page number was corrected.

L451    Page number was corrected.

L455    Page number was corrected.

L493    Number of authors was shortened to ten.

L506    Number of authors was shortened to ten.

L532    Number of authors was shortened to ten.

L542    Page number was corrected.

L560    Number of authors was shortened to ten.

L566    Number of authors was shortened to ten.

L584    Number of authors was shortened to ten.

L603    Page number was corrected.

L612    Page number was corrected.

Reviewer 2 Report

This MS is well written.  it tried to find out the factors controlling dead wood decomposition. The experiment data is fine but I think it can be improved.

My understanding, it is testing factors affecting the dead wood decomposition. It is very hard to get accurate data, such as the environment, especially the water where the dead wood is will shape the structure of the microbe population. Thus, the author should present the investigation site with images. Plus, when is the field work conducted? It affects the CO2 rate.

I would like to see how each of the factors affecting the wood decomposition by a more appealing method. Fig. 4 is not as good to get main point as a first sight. Is it hard to do some math prediction? If it is linked to the CO2 release rate along the year, it will be more eye catching. 

Author Response

Comments to reviewer #2

We have modified spell misses and unsuitable description.

This MS is well written. It tried to find out the factors controlling dead wood decomposition. The experiment data is fine but I think it can be improved.

My understanding, it is testing factors affecting the dead wood decomposition. It is very hard to get accurate data, such as the environment, especially the water where the dead wood is will shape the structure of the microbe population. Thus, (1)the author should present the investigation site with images. Plus, (2)when is the field work conducted? It affects the CO2 rate.

(3)I would like to see how each of the factors affecting the wood decomposition by a more appealing method. Fig. 4 is not as good to get main point as a first sight. Is it hard to do some math prediction? If it is linked to the CO2 release rate along the year, it will be more eye catching.

(1) We added the image of the study site as Figure 1

(2) We added the date of the measurement in L116.

(3) It is a very important point to show how each of the factors affecting the wood decomposition rate. And it seems very good idea to show CO2 release from dead wood along the year using a math prediction. However, as we showed in Figure 5, there are several factors which have cause-effect relationships. Moreover, we didn’t measure temperature and water content of dead wood through a year, which are the most important factors for the estimation of CO2 release from dead wood through a year. We just focused on the cause-effect relationships between factors in this study. Based on the results, we would like to construct the model for the estimation of decomposition rate of dead wood with different tree species, positions, and duration of decay in the next study.

We have revised our manuscript to address the other reviewers’ comments. We would like to add raw data for this study as a supplementary table for secondary use of the data. We made minor corrections as it follows.

L42      “N” was modified to “nitrogen”.

L91      “1977” was modified to “1975”.

L98      Date that measurement was conducted was added.

L100    Size (diameter and height) of the acrylic chamber was added.

L101    “Figure 1” was inserted.

L129    “degreased sample” was deleted.

L131    “degreased” was added to specify the sample condition.

L192    Figure number was modified.

L196    P value for species was modified.

L206    Figure number was modified.

L213    Figure number was modified.

L220    Name of phylum of bacteria was italicized.

L222    Figure number was modified.

L223    Figures were made bigger and rearranged. Name of phylum of bacteria in (d) was italicized.

L225    Figure number was modified.

L230    Figure number was modified.

L231    “but” was modified to “and”.

L241    Figure number was modified.

L247    Figure number was modified.

L252    Figure number was modified.

L266    Figure number was modified.

L269    “E=error variance” was deleted.

L380    Number of authors was shortened to ten.

L393    Page number was corrected.

L396    Number of authors was shortened to ten.

L398    Number of authors was shortened to ten.

L425    Doi was corrected.

L432    Page number was corrected.

L436    Page number was corrected.

L451    Page number was corrected.

L455    Page number was corrected.

L493    Number of authors was shortened to ten.

L506    Number of authors was shortened to ten.

L532    Number of authors was shortened to ten.

L542    Page number was corrected.

L560    Number of authors was shortened to ten.

L566    Number of authors was shortened to ten.

L584    Number of authors was shortened to ten.

L603    Page number was corrected.

L612    Page number was corrected.

Reviewer 3 Report

This paper determined how deadwood properties such as chemical composition, decomposer abundance, community composition, and age correlated and affected decomposition rate. The manuscript is prepared very reliably and carefully, and in my opinion this paper is worth to be published in Journal of Fungi.

Author Response

Comments to reviewer #3

We’ve checked all the sited references are relevant to this research.

We have revised our manuscript to address the other reviewers’ comments. We would like to add raw data for this study as a supplementary table for secondary use of the data. We made minor corrections as it follows.

L42      “N” was modified to “nitrogen”.

L91      “1977” was modified to “1975”.

L98      Date that measurement was conducted was added.

L100    Size (diameter and height) of the acrylic chamber was added.

L101    “Figure 1” was inserted.

L129    “degreased sample” was deleted.

L131    “degreased” was added to specify the sample condition.

L192    Figure number was modified.

L196    P value for species was modified.

L206    Figure number was modified.

L213    Figure number was modified.

L220    Name of phylum of bacteria was italicized.

L222    Figure number was modified.

L223    Figures were made bigger and rearranged. Name of phylum of bacteria in (d) was italicized.

L225    Figure number was modified.

L230    Figure number was modified.

L231    “but” was modified to “and”.

L241    Figure number was modified.

L247    Figure number was modified.

L252    Figure number was modified.

L266    Figure number was modified.

L269    “E=error variance” was deleted.

L380    Number of authors was shortened to ten.

L393    Page number was corrected.

L396    Number of authors was shortened to ten.

L398    Number of authors was shortened to ten.

L425    Doi was corrected.

L432    Page number was corrected.

L436    Page number was corrected.

L451    Page number was corrected.

L455    Page number was corrected.

L493    Number of authors was shortened to ten.

L506    Number of authors was shortened to ten.

L532    Number of authors was shortened to ten.

L542    Page number was corrected.

L560    Number of authors was shortened to ten.

L566    Number of authors was shortened to ten.

L584    Number of authors was shortened to ten.

L603    Page number was corrected.

L612    Page number was corrected.